



# GC Insights: Open R-code to communicate the impact of co-occurring natural hazards

John K. Hillier[1], Adrian Champion[2], Tom Perkins[3], Freya K. Garry[4], Hannah Bloomfield[5]

[1]Geography and Environment, Loughborough University, Loughborough, LE1 3TU, UK
[2]Aon, The Aon Centre, 122 Leadenhall Street, London, EC3V 4AN, UK
[3]Bank of England, Prudential Regulation Authority, 20 Moorgate, London, EC2R 6DA, UK
[4]Met Office, Fitzroy Road, Exeter, EX1 3PB, UK
[5]School of Engineering, Newcastle University, Newcastle upon Tyne, United Kingdom, NE1 7RU

*Correspondence to*: John K. Hillier (j.hillier@lboro.ac.uk)

**Abstract.**

Hydro-meteorological hazard is often estimated by university-based scientists using publicly funded climate models, whilst the ensuing risk quantification uses proprietary insurance sector models, which can inhibit the effective translation of risk-

related environmental science into modified practice or policy. For co-occurring hazards, this work proposes as an interim solution open R-code that deploys a metric (i.e., correlation coefficient *r*) obtainable from scientific research, usable in practice without restricted data (climate or risk) being exposed. This tool is evaluated for a worked example that estimates the impact on joint risk at an annual 1-in-200 year level of wet and windy weather in the UK co-occurring rather than being independent, and the approach can be applied to other multi-hazards and compound events in various sectors (e.g. road, rail,

telecommunications).

## Copyright Statement

## 1 Introduction

Translating scientific work into improved policy or practice is widely accepted to be desirable yet challenging (Evans, 2006; Dowling, 2015; Cordner, 2015; Margalida et al., 2015; Scott et al., 2018). For extreme weather such as flooding or wildfire (Finney et al., 2011; Berghuijs et al., 2019), known as hydro-meteorological natural hazards, two main restrictions on data



inhibit the effective translation of risk-related environmental science into modified practice in the insurance sector (e.g., Hillier and Van Meeteren, 2023). Firstly, open datasets output by publicly funded weather and climate models (e.g., from Met Office, ECMWF) are large (10s of terrabytes) yet only include selected variables. As such, they may not provide the metrics most related to extremes. Users are therefore required to have the capability to work with the data and translate it into the metrics they are interested in. Outside major projects with open licensing such as UK Climate Projections (UKCP) data might also be

released to academic researchers on non-commercial licenses. Secondly, financial risk is quantified using proprietary industry-based models and commercially sensitive data (e.g. insurance claims).

During February 2022 the storm sequence 'Dudley', 'Eunice' and 'Franklin' inflicted several hydro-meteorological hazards (snow, landslips, flooding, extreme winds) across the UK and Northwest Europe (Volonté et al., 2023a, b; Mühr et al., 2022),

resulting in multi-sector impacts (e.g. road, power distribution) and ~€3-4 billion in insured losses (Kendon, 2022; Saville, 2022). These losses illustrate the importance of considering multi-hazard risk (Kappes et al., 2012; Zscheischler and Seneviratne, 2017). In Northwest Europe, flooding and extreme wind cause the largest losses (Mitchell-Wallace et al., 2017). In wintertime, these two perils co-occur in space and time on timescales including (sub-)daily and seasonal (De Luca et al., 2017; Hillier and Dixon, 2020; Owen et al., 2021b, a), and across all timescales in between (Bloomfield et al., 2023). This

dependency exists in meteorological variables such as precipitation (e.g., Martius et al., 2016) and in impact data of losses and delays (Hillier et al., 2015, 2020). Yet, the potential for this multi-hazard relationship is not always considered in (re)insurance risk analysis; flooding and wind risk are currently modelled separately by 'catastrophe' models.

This paper presents and evaluates a transparent statistical approach (with open R-code) to combine modelled flooding and

wind risk at a seasonal timescale relevant to reinsurance. It also adds analysis with a second pair of commercial risk models to earlier results published in blogs (Hillier et al., 2023; Hadzilicos et al., 2021) that led to a modification to the Insurance Stress Tests that regulate UK insurers (Bank of England, 2022).

Research questions:


1. What is the impact of co-occurring wet and windy weather in the UK on joint risk for insurers at an annual 1-in-200 year level?
2. How useful is the proposed approach in translating scientific research into insurance industry practice?

## 2 Methods & Data

Our approach uses climate science to link risk models *via* approximations of hazard severity (Severity Indices – *SI*), such as $v^3$ for wind (Bloomfield et al., 2023; Nordhaus, 2010; Klawa and Ulbrich, 2003). Inter-hazard correlation coefficients ($r_s$)



extracted from climate models (Figure 1a) are applied to link independently derived 'catastrophe' risk models (Figure 1b) from the reinsurance industry, without exposing commercial sensitive data.

Flood-wind correlation is applied at a seasonal timescale because the annual 1-in-200 year return period (RP) loss after reinsurance (i.e. 'net' of insurance companies' own insurance) is a key metric used to calculate insurers' solvency (Hadzilicos et al., 2021). Five types of statistical method (e.g. copula) are used exactly as in the earlier analysis (Hillier et al., 2023). The calculations are detailed in the R-code provided (Supplementary material). The only inputs needed are a text file for each hazard, with one row per event.


We use data from two commercial, independently-derived risk models available from Aon. A time-series of 4,731 years is used, with flood events that have non-zero losses in the UK more frequent (~7 per year) than wind events (~3 per year), and UK-aggregated event losses are approximately log-Normal with tail end wind losses (RP > 100 years) approximately twice that of flood. For wind, correlation $\rho$ between $SI$ (sum $v^3$) and loss per event is in the range ~0.5-0.8, and about half this for
flooding ($\rho$ ~0.2-0.3, $SI$ is number of events). To test-run the R-code, events derived from the UKCP (Griffin et al., 2022; Bloomfield et al., 2023) have been created to broadly match this configuration, but are illustrative only and these outputs should not be interpreted.

To understand how the proposed approach is useful in translating scientific research into reinsurance industry practice,
statements were elicited from organisations that collaborated in the work (Bank of England, Met Office, Aon). These data are in Supplementary Material and referenced using the following format - '*The Bank of England's statements are a means of staff sharing views that challenge – or support – prevailing policy orthodoxies. The views expressed here are those of the authors, and are not necessarily those of the Bank of England or its policy committees*' [*Bank*].

## 3 Quantitative Results

Figure 1d shows the estimated effect that flood-wind co-occurrence has on annual 1-in-200 year level risk, reporting the difference between the typical assumption (i.e., independence) and a correlated case. As in the previous work the 'high' correlation, Gaussian copula case is considered most realistic, and net of reinsurance (i.e. 'after', light green) is most relevant. Lower gross, yet higher net losses are mainly caused by the flood hazard metric available. The principal result, visually synthesising results from the studies, is that uplift might be as high as ~10-14% for the very specific scenario analysed. It is
vital to realise however, that this result should not be over-interpreted, specifically should not be taken to necessarily indicate under-capitalisation of any particular firm nor of the sector in general [*Aon*].





## 4 Discussion & Reflections

Quantifying tail risk (i.e., severe but rare circumstances) is desirable for a general insurer's risk management. There are potential benefits in shared effort in addressing this complex task yet there is simultaneously the potential for commercial tensions that stem from organisations' differing roles (e.g. insurer, broker, regulator), analogous to many global industries (Ritala, 2012). Such beneficial cooperation between organisations potentially in competition has been labelled 'co-opetition' in 'paradox studies' (Gnyawali and He, 2008; Brandenburger and Nalebuff, 1996; Smith et al., 2017). Various ways of handling this exist (e.g., Stadtler and Van Wassenhove, 2016) although a critical part can be succinctly summarised as '*Partnerships require good networks, time, and trust to develop*' [*Met Office*]. The collaboration (Bank of England, Aon, Met Office, Loughborough University, Verisk) from 2018-2023 that developed the open R-code is seen as a successful example of a co-opetition project (Hillier and Van Meeteren, 2023). The finding that co-occurrence might plausibly raise annual joint UK flood-wind losses net of reinsurance by up to ~10% (Figure 1d) is only applicable in this particular analysis but as an indicator that correlated hazards are worth considering it is seen as a valuable contribution [*Aon, Met Office, Bank*], and developing the open source R-code 'tool' is also considered a benefit:

'*An important step in bringing together publicly funded climate model data and industry-based modelling in a transparent way*' [*Aon*]

But, to what extent is it useful? It is a valuable first step [*Aon*], which can be an informative tool for those insurers who haven't previously captured these dependencies because it is prudent to explore rather than ignore potential dependencies [*Bank*](Hillier et al., 2023). The approach can be applied to sets of events (e.g., flood, storm) within existing models [*Aon*], making it quickly implementable. However, whilst accounting for uncertainty in some choices (i.e., dependency structure, hazard correlation) there are many other variables (e.g., reinsurance structure) that are only partially mitigated by using multiple inputs (e.g., different commercial risk models). Hence, critically, careful interpretive judgement is needed:

'*Where tools such as this R-code are applied to inform a view of risk, caveats and assumptions should be considered; users should be satisfied a tool is being used in appropriate circumstances.*' [Bank]

Illustratively, it would be unwise to apply the headline result to an insurer's unique portfolio. For this reason 'open-source' and 'transparent' are highlighted as key benefits of this approach [*Aon, Met Office*]. Possible applications include exploring the sensitivity of peril co-occurrence to different financial structures (e.g. number of reinstatements). [*Bank*]. As independent and open source it also provides a means to benchmark future similar work in this field [*Aon*], and can be applied to different hazards.





Overall, we conclude that our approach is one useful interim solution prior to, and perhaps justifying, more extensive modelling. It is a bridge, deploying a metric (i.e., correlation coefficient) obtainable from scientific research, which is usable in practice without restricted data being exposed. More generally, it is an example of embedding environmental science in to practice and policy by identifying a simple, pragmatic means (e.g., *r*) of estimating the impact on a critical industry-relevant metric, and paves the way for similar methods to be applied within other sectors (e.g. rail, road, power distribution,
telecommunications).

**Figure 1 – Pathway from hazard to impact on loss. (a)** Wintertime correlation, Spearman's Rank ($r_s$) within various time window lengths for the Oct-Mar season, between flood proxies and extreme wind in Great Britain (Bloomfield et al., 2023). Rain (purple)
and river flow (yellow) are related to wind hazard for climate model (UKCP, solid lines) and historical (ERA5, dashed) and rain (purple). Error bands are 95% confidence. **(b)** Illustration of the method, i.e. statistically linking two independent risk models (red / blue) via their hazard proxies. **(c)** Illustrative exceedance probability curves for correlated (dark grey) and independent (light grey) cases, the difference between which is the effect of co-occurrence, with the 1-in-200 return period (*p* = 0.005) of particular interest in insurance **(d)** Indicative impact of a correlation between flooding and wind hazards on annual losses for the whole UK market at
a 1-in-200 year return period. Box plots display the distribution created by five types of correlation (e.g. copula – see b). As in Hillier et al (2023) the Gaussian is 'best' and highlighted (black dot) as it best fits annualized hazard data at Site W of Hillier & Dixon (2020), with open circles from that 2023 study.



## 5 Ethics Statement

Ethical approval was given by the Ethics Review Sub-Committee at Loughborough University.

## 6 Conflicts of interest

JH is an Executive Editor of *Geoscience Communication*. The authors have no other competing interests to declare.

## 7 Author contributions

JH conceived the work and undertook the analysis. All authors contributed to the drafting, writing and review of the
manuscript.

## 8 Data & Code Availability

The R-code used is openly available, and is in supplementary material to this article, along with guidance and a worked example with idealised data. Data from the proprietary insurance sector models used are not available.

## 9 Acknowledgements

JH was funded by NERC (UKRI) Knowledge Exchange Fellowships NE/R014361/1 and NE/V018698/1. We are grateful to Aon, Bank of England and the Met Office for their collaboration.

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
