# Peer review of "GC Insights: Open R-code to communicate the impact of co-occurring natural hazards"

_EGUsphere, 2023_

## Author Comment (AC1)

**Response to reviewer comments**

We thank both reviewers for their constructive comments. Please find below a detailed response. Reviewers' comments are in grey, and responses in black. Whilst a revision isn't required by *GC* at this stage, we present it in this style as we found it easiest to understand and work through the comments this way (e.g. seeing what it's possible to do within the word limit).

Importantly, the Reviewers' comments have help to understand where we have relied on implicit or assumed knowledge too much. So, we are happy to modify the text improve its readability and to widen its potential audience.

**Reviewer 1**

R1.1 - The paper demonstrates a pragmatic approach to the combination of single-hazard losses. It is laudable to see that the underlying code is open-source and freely available.
> Thank you.

R1.2 - The paper would have benefited from reviewing existing literature of compounding hazards beyond cited Zscheischler (where I would rather suggest to cite https://doi.org/10.1038/s41558-018-0156-3), e.g. https://zenodo.org/records/7135138, https://doi.org/10.5194/esd-7-659-2016, https://doi.org/10.5194/nhess-22-1487-2022 and https://doi.org/10.1016/j.jenvman.2015.11.011 (most findings also hold rue w/o climate change) and to (if possible) include latest contributions based on event loss tables (e.g. https://eartharxiv.org/repository/view/5286), approaches quite close to the one presented here.
> Thank you. We have swapped the Zscheischler reference for the 2018 one you suggest, and added Ward (2022). We disagree that the Stalhandske (2024) paper is best described as close to the work here – it is a rare example of full joint modelling from an underlying climate model. One of the purposes of the R-code here is as a scoping exercise to determine if such extensive modelling is worth the effort. Instead we include Stalhandske (2024) in the slightly expanded context given in the Introduction, although this use of words precludes any wider review of the compound hazards literature.

R1.3 - In addition to what the paper covers, where r is determined from UKCP 'today', what about climate change? You could determine the r for different GCMs under select RCPs at time-horizons.... and possibly contribute to physical risk disclosure. Might be worth a few sentences at least as an outlook.
> Thank you for this suggestion, it is in line with comment R2.6. I have added a few words on future climate to the end of the abstract and the end of the main text. I was not possible, however, to go into more detail about where different *r* value estimates might come from (e.g. GCMs, and RCPs at varied future times). Interestingly, some of my work nearing completion indicates that the dependency structure may not change, even as the individual hazards vary in frequency.

In detail, a few minor points:

R1.4 - line 50: ..analysis with a second pair of commercial risk models… 'second' is hard to understand here. It becomes clear once one looks at Figure 1b.
> It is difficult to explain the sequence of projects concisely and, since it is not the critical point here, this sentence has been removed.

R1.5 - line 91: [AON] at the end of the sentence. You state above that the loss table stems from a model run by AON. AON here in brackets cannot be a citation, and with above statement(s) of caution, I do not see any need to reference AON here - what does it exactly stand for? Could it be you only missed to put it in *italic* font, so it would be a quote as you explain in lines 81-83?
> You are correct. The quoted fragment is now in *italic*.

R1.6 - line 96: Co-opetition. UK's Flood Re is a good example of a public-private-partnership solution that also bears fruits to all market competitors. Might be worth mentioning (especially as you use flood as a demonstrating hazard)
> Thank you for the suggestion.  However, Flood Re is an example of a government created and backed reinsurance pool, rather than co-opetition between organisations whose relationships hold implicit tensions.

R1.7 - line 104, [AON, Reto Office, Bank]: Again, what does this imply, as it is not in italic, hence does not look like a quote? Is it the entities that endorsed the statement? Did others (e.g. Verisk) not? Would it be an option to clarify this upfront (above, line 81ff, where you make the general statement, add that […] means endorsement of a statement) and the reader would then know how to interpret these listings.
> Thank you. This is a good point. In the Methods I have clarified that these references e.g. [*Aon*] apply to either quotes or paraphrased views that can be directly drawn from the statements the companies provided (in supplementary material, so the reader can examine and agree or disagree with any interpretation).

R1.8 - And as a minor detail, should it not be Bank of England throughout the text (or you state first time that you will abbreviate).
> Thank you.  We only abbreviate in the references, i.e.  [*Bank*], and believe that doing so for the first quote where the Bank's name is used in full in the quote is sufficient warning that we will abbreviate in this way. We refrain from adding any words to retain the space for other changes, within the tight 1500 word limit.

R1.9 - line 116/117: This statement holds for any tool
> Indeed. Thank you for noting this.  However, the Bank of England in their role as co-authors were keen that it be pointed out explicitly for this tool.

**Reviewer 2**

R2.1 - In this manuscript, the authors present an open R code that can transparently translate projected co-occurring natural climate-related hazards (illustrated with wind-flood events) into financial risk quantification for the reinsurance sector. This is an interesting manuscript and it makes an original contribution to the literature by tackling the important theme of open science-policy-practice. Further refining the objectives and methods will significantly enhance the presentation of this work. I hope you find the following comments useful in revising your manuscript for publication.
> Thank you. We will take your comments, and endeavour to improve the manuscript within the 1500 word limit of the *GC Insights* format.

General comments:

R2.2 - I find that the title doesn't fully reflect the content of the paper. It could be enhanced by adding the keyword "(financial) risk", and by changing "communicate" (which is very broad) to "translating" (as used on L14).
> Thank you.  We have modified the title in line with your suggestions.

R2.3 - Who are the intended users of the code you developed? I think that it would be great to clarify this in the manuscript, perhaps already in the introduction. It would appear that the primary users are people from the reinsurance sector, but I am wondering if your code could also be used by researchers, perhaps in collaboration with partners from the reinsurance sector, further fostering collaborations. Out of curiosity, do you have some information you could share on the partners' perspectives on "anyone" (in theory) being able to use your code to calculate future financial risk?
> The breadth of intended users is now clarified in the Introduction.
> On the second part of your question, we're afraid that we don't have any information that we can share in a means attributable to the partners. The statements in the Supplementary Material were cleared by both management and legal departments of the respective organisations, and so seeking to modify or expand these is not in the scope of the current work.

R2.4 - Could you please clarify throughout the manuscript what "flood risk" refers to. Do you use "flood proxies" as mentioned in the caption of Fig. 1 or are floods defined using a different method?
> Risk throughout has now been clarified to mean modelled financial risk (i.e. monetary loss), whilst for consistency and ease of reading 'proxies' have been replaced throughout with 'severity indices' (*SI*). *SI* are a type of proxy (Hillier and Dixon, 2020).

R2.5 How are the ensembles and error bands exemplified in Fig. 1a and 1c used to generate Fig. 1d? Is the median/mean used only?
> Only single values are used, with no error propagation in this sense. This is done pragmatically as in the analysis for Bank of England blog this manuscript adds some transparency to (i.e., Hillier et al., 2023). We have altered the Methods section to clarify that two *r* values for high and low dependency cases are taken, adding to the annotation in Fig 1d. Also, annotation on the black arrow in Fig 1c now states that only the means are taken.

R2.6 - This leads me to wonder what exceedance probability curves might look like for future climate projections where the ensemble might cover a larger spectrum of possibilities. What would be the implications on this method if the uncertainty is so large that using the projected ensemble median/mean does not fully capture the broad spectrum of possibilities for financial risk information? A small reflection on this point in the manuscript would be great.
> Interestingly, some of our work nearing completion indicates that the dependency structure may not change, even as the individual hazards vary in frequency. We have re-read and re-worded the statement briefly reflecting on this in the discussion. However, our priority was setting the context for the work better in the introduction (as you suggest), so have not the space to expand this point.

R2.7 - To what extent is the methodology applicable to other regions of the world, different climate model outputs, and/or different hazards? Could you please give readers a brief overview of users' level of involvement/changes they will need to make to the code to run it for a different case?
> Assuming they can get the 3-column input files and a correlation coefficient, no changes are needed to the body of the R script to apply it to different hazards, regions, climate models. A clarification has been added to the methods section.

Specific comments:

R2.8 - L13: Could "university-based scientists" be changed to "researchers" more generally, to include scientists working at research centres that are not academic institutes?
> Done

R2.9 - L14: Please Specify that you are referring to financial "risk" (vs. other forms of risk).
> Changed on L18 to clarify the example used. The overall principle could apply to any form of quantifiable risk.

R2.10 - L15: If the word count permits, it would be great if you could briefly contextualize the focus on co-occurring hazards.
> With the abstract limited to 3 sentences in *GC Insights*, we're afraid that we cannot see how to add further contextualization here.

R2.11 - L15-16: As this is an "interim solution", could you please share some information in the manuscript on what the longer-term goal/tool might look like?
> The longer-term full solution is to model hazards and risk jointly from the same underlying climate model. This has very recently been done for Tropical Cyclones (Stalhandske et al., 2024; Verrisk, 2024), but not for any other hazards I am aware of. This is a significant effort, and users are interested to know if it is likely to be a substantial effect before investing so heavily. This information has been added briefly to the Introduction.

R2.12 - L16: Please specify what the correlation coefficient is calculated on.

> Clarification that the correlation coefficient is between hazards (i.e. not risks) has been added.

R2.13 - L28-29: "For extreme weather such as flooding or wildfire, […], known as hydro-meteorological natural hazards". Perhaps a more accurate rephrasing could be: "For hydro-meteorological natural hazards, such as flooding or wildfire, caused by extreme weather".
> Changed. The original phraseology was designed for those who are not environmental scientists.

R2.14 - L31-33: How does your work tackle the issue related to large climate datasets that might not contain the necessary variables?
> Thank you for prompting me to clarify this. The large climate datasets are used by the risk model creators, who can then provide the necessary impact-related metrics (e.g. storm severity indices - SI) on an event basis, data that are smaller in quantity and easier to handle. L31-33 have been edited to simplify and clarify them, and the first paragraph of Methods has been altered to make it clearer that the *SI* metrics relevant to extremes are available within the risk models.

R2.15 - L44-46: "This dependency exists in meteorological variables such as precipitation (e.g., Martius et al., 2016) and in impact data of losses and delays". Can you please clarify this sentence? It is not clear what "dependency" you are referring to and what "losses and delays" means.
> The co-occurrence referred to in the previous sentence causes the dependency, with dependence being the term preferred by statisticians (i.e. preferred over correlation). I have simplified the previous sentence with the intention of making the link clearer. The sentence has been amended to clarify what the losses and delays are.

R2.16 - L47: Is this the case everywhere in general or do you know of any compound risk models that you could reference here?
> Globally, for all hazards, tropical cyclone is a recent exception with 2-3 commercial models and one recent academic paper (Stalhandske et al., 2024). This is now included. I do not mention 'secondary' perils, such as fire following earthquake, in models here as they are joined by statistical methods (I suspect) in the proprietary commercial products.

R2.17 - L49-50: Related to the above, do you know if there are any other transparent codes for this type of work (and if so how your approach differs), or if this is the first of its kind? It would be some great context to have for readers less familiar with the literature.
> Copula packages exist in various languages, and could be used to create a R script similar to the code here, but as far as we are aware they have not been applied to this purpose (i.e. joining models at the hazard level to estimate impact on joint risks). Commercial products (e.g. Remetrica) can also be adapted to do this, and were for the first iteration of this project (Hadzilicos et al., 2021), but nothing open. I've clarified at the end of the introduction that we believe this is the first code for this statistically-based approach to linking this sort of model.

R2.18 - L50-52: "It also adds analysis with a second pair of commercial risk models to earlier results published in blogs […]". I feel like this second part needs a bit of introduction.
> This reference to the first iteration of the project in 2021 (Hadzilicos et al., 2021) has been removed to create space for greater context (e.g. R2.11, R2.16, R2.17).

R2.19 - L56-57: It would be great if you could please clarify that "1-in-200 year level refer" refers to monetary losses here.
> Research question modified with the intention of doing this.

R2.20 - L61-62: What kind of model are the simulated flood events from? i.e., is it a hydro-meteorological model that takes into account flood generating mechanisms such as rainfall and groundwater, but indeed not wind? I guess what I'm getting at is whether those events already account for wind if they're based on a coupled ocean-atmosphere-land model.

> We've clarified in Methods that these are fluvial flood events. We are not permitted to say which commercial models were used, but I can confirm that they conform to sector norms see Chapter 3 of Mitchell-Wallace *et al.* (2017) in that flooding is considered entirely separately to wind.  In the Introduction, I've clarified how the hazards are modelled (i.e. entirely separately) by referenced the textbook (Mitchell-Wallace et al., 2017) to indicate how flooding and wind models work in this sector.  This is all the word limit permits.

R2.21 - L65-66: The link between seasonal correlations and the 1-in-200 year return period is not clear to me. Please clarify in the manuscript.
> Thank you.  I think the word order here made the sentence unclear. It has been changed. The point here is the use of the correlation value for the longest timescale (whole winter, seasonal) as this is the closest to the financial annual assessment used for insurers' solvency.

R2.22 - L67: Please clarify briefly in the text what these five statistical methods are used for. I can only see four methods mentioned in Fig. 1b. Is there one missing?
> Changed. I was counting the two t-copulas (different degrees of freedom) as two methods. I can see how it would be clearer to say four.

R2.23 - L68-69: Could you clarify a bit further in the manuscript what input data (e.g., dates, magnitude, monetary losses?) are needed from the user?
> This has been added.

R2.24 - L75: I don't understand how a correlation can be calculated between the number of events and the loss per flood event. Are the flood events ranked and correlated to the losses they incurred?
> Apologies. This was a typo. The correlation is per year, and 'event' has been changed to 'year'.

R2.25 - L87: How is the Gaussian copula case assessed to be most realistic? It would be great to have a bit more information on this in the manuscript.
> As it says in the figure caption the Gaussian is thought to be 'best' in Hillier et al (2023) as it best fits annualized hazard data at Site W of Hillier & Dixon (2020). This was a pragmatic decision based on fitting a range of copulas to these data in a 'real-world' workflow. Bringing that decision into this paper allow us to fulfil one of the purposes of a *GC Insights* article i.e. to bring transparency – for the numerical elements at least - to an industry blog (Hillier, 2023) https://bankunderground.co.uk/2023/04/13/what-if-its-a-perfect-storm-stronger-evidence-that-insurers-should-account-for-co-occurring-weather-hazards/.  I can't add information on any internal discussions behind this due to confidentiality. However, as expressed in the Bank Underground article, understanding the variability and range of outcomes is important.  For these reasons we will not elaborate further on this choice.

R2.26 - Fig. 1: a) The right "0.4 'low' case" y-label is not aligned with the left 0.4 y-label.
> Fixed. And, some annotation on Fig. 1b has been added for clarity e.g. Gaussian is now 'Gaussian copula'

d) It is not clear to me how the data is split into different boxplots with low and high correlations. Could you explain this briefly in the text/caption?
> Understood.  The explanation in the figure caption has been expanded.

Hadzilicos, G., Li, R., Harrington, P., Latchman, S., Hillier, J. K., Dixon, R., New, C., Alabaster, A., and Tsapko, T.: It's windy when it's wet: why UK insurers may need to reassess their modelling assumptions, Bank Underground, 2021.

Hillier, J. K. and Dixon, R.: Seasonal impact-based mapping of compound hazards, Env. Res. Lett., 15, 114013, https://doi.org/10.1088/1748-9326/abbc3d, 2020.

Hillier, J. K., Perkins, T., Li, R., Bloomfield, H., Lau, J., Claus, S., Harrington, P., Latchman, S., and Humphry, D.: What if it's a perfect storm? Stronger evidence that insurers should account for co-occurring weather hazards, Bank Underground, 2023.

Mitchell-Wallace, K., Jones, M., Hillier, J. K., and Foote, M.: Natural Catastrophe Risk Management and Modelling: A Practitioner's Guide, Wiley, Oxford, UK, 506 pp., 2017.

Stalhandske, Z., Steinmann, C. B., Meiler, S., Sauer, I. J., Vogt, T., Bresch, D. N., and Kropf, C. M.: Global multi-hazard risk assessment in a changing climate, Scientific Reports, 14, 5875, https://doi.org/10.1038/s41598-024-55775-2, 2024.

Verrisk: https://www.verisk.com/en-gb/products/tropical-cyclone-models/, last access: 20 June 2024.